# Prevalence of Sarcopenia in Knee Osteoarthritis: A Systematic Review and Meta-Analysis

**DOI:** 10.3390/jcm12041532

**Published:** 2023-02-15

**Authors:** Francesco Pegreffi, Alice Balestra, Orazio De Lucia, Lee Smith, Mario Barbagallo, Nicola Veronese

**Affiliations:** 1Dipartimento di Scienze per la Qualità della Vita–QUVI, Università di Bologna, 47921 Rimini, Italy; 2Unit of Clinical Rheumatology, Department of Rheumatology and Medical Sciences, ASST Centro Traumatologico Ortopedico G. Pini-CTO, 20122 Milan, Italy; 3Centre for Health, Performance and Wellbeing, Anglia Ruskin University, Cambridge CB1 1PT, UK; 4Department of Internal Medicine, Geriatrics Section, University of Palermo, 90128 Palermo, Italy

**Keywords:** sarcopenia, knee osteoarthritis, meta-analysis

## Abstract

An association between knee osteoarthritis (OA) and sarcopenia has been proposed, but the evidence is controversial, with the recent literature showing disparate results. Therefore, we aimed to perform a systematic review and meta-analysis to evaluate the prevalence of sarcopenia in knee OA patients compared to people not affected by this condition. We searched several databases until 22 February 2022. The data regarding prevalence were summarized using odds ratios (ORs) with their 95% confidence intervals (CIs). Among the 504 papers initially screened, 4 were included for a total of 7495 participants with a mean age of 68.4 years, who were mainly females (72.4%). The prevalence of sarcopenia in people with knee OA was 45.2%, whilst, in the controls, it was 31.2%. Pooling the data of the studies included that the prevalence of sarcopenia in knee OA was more than two times higher than in the control group (OR = 2.07; 95%CI: 1.43–3.00; I^2^ = 85%). This outcome did not suffer any publication bias. However, after removing an outlier study, the recalculated OR was 1.88. In conclusion, the presence of sarcopenia in knee OA patients was high, affecting one person in every two persons and was higher than in the control groups included.

## 1. Introduction

Sarcopenia remains one of the most widespread geriatric syndromes and the major leading cause of mortality [1]. The principal physiological substrate underlying this disease comprises the process of an age-related decline in muscle strength and a degenerative loss of skeletal muscle mass and function [2]. Sarcopenia represents an impaired state of health in terms of functional decline and mobility disorders [3], increased risk of falls and fractures [4], loss of independence, impaired ability to perform activities of daily living and disabilities [5], frailty and, finally, mortality [6].

Osteoarthritis (OA), by its turn, is the most common articular disorder, rising as the population ages and affecting up to a fifth of people worldwide, thus being the most prevalent cause of joint disease globally [7]. Although its pathophysiology remains unclear, the adverse extra-articular health outcomes leading to progressive disability over time, such as cardiovascular mortality, falls, and conditions associated with subclinical atherosclerosis, were previously described [8].

Being that OA and sarcopenia are very frequent conditions, it is not unexpected that they often co-exist, raising the possibility of them interacting with one another. The biomechanical effect resulting from an altered bone and periarticular muscle cross-talk could be responsible for the atrophy or weakness of the muscles themselves, thus leading to the development, progression and severity of OA [9]. The interplay between OA and sarcopenia is plural and hardly straightforward, stemming from a shared mechanism of chronic low-grade inflammation, on its origin, to a superimposable risk of multiple contributing factors and lifestyle [9].

Despite the multiple interconnections postulated to explain a possible modulating effect of OA on sarcopenia, data demonstrating a relationship between the two diseases remains elusive and, at times, conflicting. Taking these considerations into account, we aimed to perform a systematic review and meta-analysis to evaluate the prevalence of sarcopenia in knee OA patients, compared to people not affected by this condition, and whether it could be demonstrated as significant evidence concerning an association between sarcopenia and knee OA in terms of the latter to OA development, progression and severity.

## 2. Materials and Methods

This systematic review adhered to the PRISMA statement [10] and followed a pre-planned but unpublished protocol that can be requested by contacting the corresponding author.

### 2.1. Data Sources and Searches

Two investigators independently conducted a literature search using PubMed/Medline, Scopus and Web of Science from their database inception until 22 February 2022 and included observational studies investigating the presence of sarcopenia in patients affected by knee OA. In PubMed/Medline, the following search strategy was used: (Osteoarthriti* OR osteo arthritis OA or osteo-arthriti* OR osteoarthros* OR osteo-arthros* OR osteoarthros* OR arthrosis) AND (sarcopenia)”, adapting the search for Scopus and Web of Science. Any inconsistencies during the title, abstract and, finally, full-text screening were resolved by consensus with a third senior author (NV). 

### 2.2. Study Selection

The inclusion criteria for this meta-analysis were as follows: (i) an observational study; (ii) reporting a diagnosis of knee OA, using radiological and/or medical information; (iii) reporting data on sarcopenia, independently from the definition used (e.g., low muscle mass, low physical performance or a combination); (iv) reporting data regarding the controls defined as people without any evidence of knee OA. Studies were excluded if: (i) they did not include humans; (ii) they did not include a control group, i.e., without knee OA.

### 2.3. Data Extraction

Two independent investigators extracted key data from the included articles on a standardized Excel spreadsheet, with a third independent investigator (NV) checking the data. For each article, we extracted data on the authors’ names, year of publication, country, sample size, age and percentage of females, and the mean body mass index (BMI) diagnostic criteria used for the definition of knee OA and sarcopenia.

### 2.4. Outcomes

The primary outcome was the prevalence of sarcopenia in knee OA versus the controls.

### 2.5. Quality Assessment

Two independent authors (NV, FP) carried out the assessment of the studies’ quality using the Newcastle–Ottawa Scale (NOS) [11]. The NOS assigns a maximum of 9 points based on three quality parameters: selection, comparability and outcome. As per the NOS grading in past reviews, we graded studies as having a high (<5 stars), moderate (5–7 stars) or low risk of bias (≥8 stars) [12].

### 2.6. Data Synthesis and Analysis

The primary analysis compared the cumulative prevalence of sarcopenia between participants with knee OA and participants without this condition. We calculated the odds ratio (OR) with their 95% confidence intervals (CIs), applying a random-effect model [13].

Heterogeneity across studies was assessed by the I^2^ metric and χ^2^ statistics. Given the significant heterogeneity (I^2^ ≥ 50%, *p* < 0.05), we conducted a series of meta-regression analyses [14], using as moderators the mean age, percentage of females, and mean BMI. To explain the heterogeneity of the results, a sensitivity analysis using the one-study removed method was conducted. Publication bias was assessed by visually inspecting the funnel plots and using the Egger bias test [15].

For all analyses, a *p*-value less than 0.05 was considered statistically significant. All analyses were performed using STATA, version 14.0 (StataCorp, College Station, TX, USA), and RevMan 5.4 (Revman International Inc., New York, NY, USA).

## 3. Results

### 3.1. Literature Search

As shown in Figure 1, we initially screened 504 articles. Among the 56 evaluated through the full-text examination, mainly removed since they did not report a control group, four studies were finally included [16,17,18,19].

### 3.2. Descriptive Characteristics

As shown in Table 1, the four cross-sectional studies included 7495 participants with a mean age of 68.4 years and were mainly females (72.4%). The mean BMI was 24.4 kg/m^2^. Three studies were conducted in Asia, whilst the other one was in Europe. Three studies used a definition of knee OA with only radiological information and with a moderate severity according to the Kellgren–Lawrence scale. The diagnosis of sarcopenia was made in two studies using only low muscle mass, whilst the other two used a combination of low muscle mass and physical performance. Finally, the quality of the studies was, in general, very good, without any study presenting a high risk of bias. 

### 3.3. Prevalence of Sarcopenia in Knee Osteoarthritis

As shown in Figure 2, the prevalence of sarcopenia in people with knee OA was 45.2%, whilst, in the controls, it was 31.2%. Pooling the data of the four studies included that the prevalence of sarcopenia in knee OA was more than two times higher than in the control group (OR = 2.07; 95%CI: 1.43–3.00; *p* = 0.0001; I^2^ = 85%). This outcome did not suffer from any publication bias according to the funnel plot inspection and Egger’s test (−0.46 ± 1.41; *p* = 0.76). 

### 3.4. Sensitivity and Meta-Regression Analyses

Since the outcome of our interest was characterized by high heterogeneity, we used a sensitivity analysis, removing one study at each step. After removing the cohort of females of Sung et al., the heterogeneity significantly decreased to an I^2^ = 20%, with a recalculated OR of 1.88 (95%CI: 1.53–2.30). On the contrary, the age (*p* = 0.59), percentage of females (*p* = 0.48) or BMI (*p* = 0.06) did not moderate our findings. 

## 4. Discussion

This systematic review with the meta-analysis represents a step forward in the thus far investigated interplay between two major geriatric clinical entities that, when associated, could progressively lead to devastating consequences. When pooling the data of four selected studies, evaluating a sample of 7495 subjects, we clearly identified that the prevalence of sarcopenia in people with knee OA (45.2%) resulted more than two times higher than in the control group (31.2%) since the OR was 2.07. 

A complex interaction of multiple factors contributes to the development and progression of OA, making physiopathology harder to define. Initially, biomechanics and macroscopic structural factors regulating bone alignment and cartilage kinetics were the main focus. Moreover, in the latter years, researchers focused the lens on biomechanics, interpreting the factors that contribute to cartilage destruction and the remodeling of subchondral bone [20]. Upper extremities and lower extremities OA, due to the distinct biomechanics and presence of a genetic background, could be considered different entities [21].

Furthermore, even confronting weight-bearing joints, such as the knee, the recognition of inflammatory damage patterns activating the complement cascade as pathogenetic factors highlights the concept that OA may not actually be a single disease [22]. In parallel to the shift from macro- to micro-architecture key-defining events leading to bone and cartilage dysfunction, intra-articular and extra-articular aspects have been increasing in importance in OA management [23]. The latter, recently, was demonstrated to play a crucial role in facilitating and perpetuating knee OA’s beginning and progression. Joint space narrowing and periarticular muscle weakness are considered risk factors that predispose to knee pain and disability [24].

In our systematic review, knee OA was diagnosed and classified according to the radiographic criteria of the Kellgren and Lawrence classification as moderate severity, and in one study, the clinical information was integrated. Furthermore, sarcopenia was defined as low muscle mass and/or physical performance, as suggested by the EWGSOP criteria. In line with both these considerations and the previous scientific literature postulating that OA and sarcopenia may be co-existing conditions—an entity usually called osteosarcopenia [25]—our work aimed to study the prevalence between sarcopenia and knee OA because the relationship between these two clinical entities is still unclear and no strong consensus has been reached [26].

In knee OA, quadriceps weakness is an issue of paramount importance. The previous literature describes subjects with OA of the knee who are unable to effect a maximal voluntary contraction of the quadriceps [27]. This phenomenon could be attributed to arthrogenous muscle inhibition due to altered afferent input from the diseased joint and a consequent reduction in efferent motor neuron stimulation of the quadriceps or to the high prevalence of sarcopenia, as reported in our work. Another issue supporting the aforementioned observation is that the prevalence of sarcopenia was higher in 85-year-olds compared to 70-year-olds, ranging from 42–62%, as reported by a cohort study [28]. It is particularly remarkable that in our review, sarcopenia associated with knee OA was superimposable (45.2%), but the mean age was 68.4. We can hypothesize that the association of the two geriatric syndromes is responsible for an early onset of sarcopenia.

A large longitudinal cohort study found that, in 1653 subjects without radiographic knee OA at baseline, an increased risk of radiographic OA was not associated with sarcopenia alone but rather with sarcopenic and body composition-based obesity [29]. In our review, the mean BMI was 24.4, which was at the limit between the normal and overweight values. Thus, excluding sarcopenic obesity, a metabolic-driven situation in between obesity and sarcopenia, affects our review’s data, thus giving a real description of the coexistence of the two geriatric syndromes. 

Since the outcome of our interest was characterized by high heterogeneity, we used a sensitivity analysis, removing one study at each step. After removing the cohort of females of Sung et al., the heterogeneity significantly decreased to an I^2^ = 20%, with a recalculated OR of 1.88 (95%CI: 1.53–2.30), maybe indicating a different role of gender in the association between sarcopenia and knee OA.

The findings of our systematic review must be interpreted within its limitations. First, the studies that were included were observational and cross-sectional/case-control: therefore, a high probability of reverse causation, i.e., people with sarcopenia have a higher prevalence of knee OA, is possible. Second, the outcomes of our interest are characterized by high heterogeneity, even though it is mainly driven by one study. Third, different definitions of sarcopenia were used in this systematic review, including definitions using only muscle mass assessment. In this regard, several societies indicated that low muscle mass identifies only a stage of pre-sarcopenia than sarcopenia itself, for which other measurements are needed, such as physical performance or muscle strength. [2,30] In our systematic review, however, two studies reported the data by combining information regarding low muscle mass and low physical performance and another two as only low muscle mass, and this choice could create an important bias in our findings. However, our findings reflect common situations in the meta-research regarding sarcopenia since, particularly in older works, only one domain of sarcopenia was assessed. Finally, the large majority of studies were from Asia, having, consequently, poor applicability to other contexts. 

## 5. Conclusions

The presence of sarcopenia in knee OA in patients was high, affecting one person on two and was higher than in the control groups included. Future studies with a longitudinal design are required to confirm whether the incidence of sarcopenia will be higher in people affected by knee OA and the role of some important mediators, such as obesity, in this association. 

## Figures and Tables

**Figure 1 jcm-12-01532-f001:**
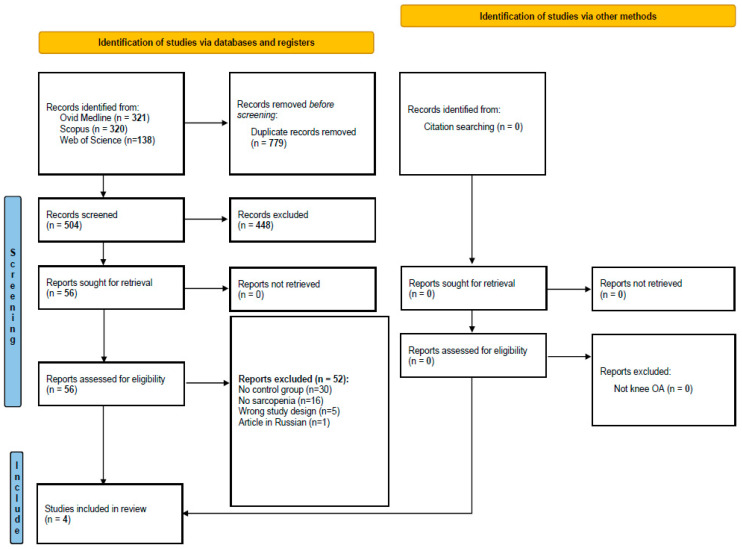
PRISMA flowchart.

**Figure 2 jcm-12-01532-f002:**
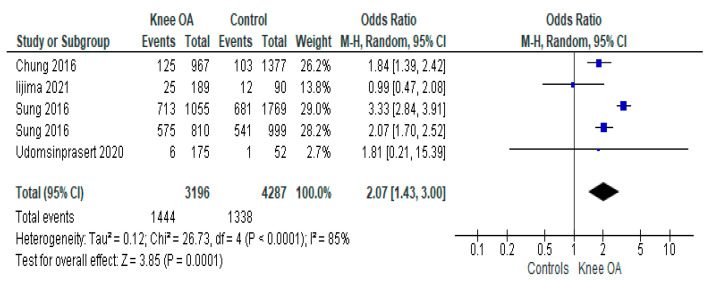
Prevalence of sarcopenia in patients with knee osteoarthritis versus controls in cross-sectional studies [16,17,18,19].

**Table 1 jcm-12-01532-t001:** Descriptive characteristics of the studies included.

Author of Study	Type of Study	Total Sample	Mean Age	% of Females	Mean BMI	Knee OA Definition	Knee OA Severity	Sarcopenia Definition	Quality
Chung, 2016 [16]	cross-sectional	2344	62.9	57.5	24.1	Only radiological	Moderate	Only low muscle mass	9
Iijima, 2021 [17]	cross-sectional	291	72.7	78.7	24.1	Only radiological	Moderate	Both low MM and physical performance	9
Sung Jin, 2016 [18]	cross-sectional	4633	72.6	56.5	24.2	Only radiological	Moderate	Only low muscle mass	8
Udomsinprasert, 2020 [19]	cross-sectional	227	65.5	98.2	25	Combination of radiological and clinical information	Not specified	Both low MM and physical performance	7
**Total**	**4 studies cross-sectional**	**7495**	**68.4**	**72.7**	**24.4**	**3 studies: only radiological; 1: combination**	**3 studies: moderate severity; 1: not specified**	**2 studies: only low muscle mass; 2 studies: both low muscle mass and physical performance**	**Median = 9**

## Data Availability

Data are available upon reasonable request to the corresponding author.

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
