# Peer review of "Prevalence of Sarcopenia in Knee Osteoarthritis: A Systematic Review and Meta-Analysis"

_jcm, 2023, doi:10.3390/jcm12041532_

Round 1

Reviewer 1 Report

General Comments

The authors performed a systematic review and meta-analysis to evaluate the prevalence of sarcopenia in knee OA patients, compared to people not affected by this condition. They found that the prevalence of sarcopenia in people with knee OA (45.2%) resulting more than two times higher than in the control group (31.2%) from the data of the four selected cross-sectional studies. They concluded that the presence of sarcopenia in knee OA in patients was high, affecting one person over two and being higher than the control groups included.

Although the studies included were observational and cross-sectional/case-control, this could be important baseline data for future studies with a longitudinal design.

Author Response

R: We would like to sincerely thank the Reviewer 1 for the positive comments. We have tried to further improve the work using the comments of the Reviewers 2 and 3.

Reviewer 2 Report

February 2, 2023

Dear Editor

Thank you for the invitation to review the manuscript entitled Prevalence of Sarcopenia in Knee Osteoarthritis: a Systematic Review and Meta-analysis" for the Journal of Clinical Medicine. This study was a well-written article. I appreciate the steps in the method for systematic review and meta-analysis that are appropriately presented. However, there are some questions as follows

  1. Studies that used only low muscle mass could not be diagnosed sarcopenia. Could it be only pre-sarcopenia?
  2. Who is the control for this study? Please clarify.
  3. The authors stated that the prevalence of sarcopenia in knee OA was 45.2% while in control was 31.2%. How can it result in the prevalence of sarcopenia in knee OA being more than two times higher than in the control group? Even OR = 2.07, can it be interpreted as presented?
  4. In addition, after sensitivity analysis to decrease heterogeneity, the recalculated OR was lessened (1.88). The reviewer suggested adding this in the abstract.

              Thank you very much for your invitation.

Best Regards,

Author Response

Dear Editor

Thank you for the invitation to review the manuscript entitled “Prevalence of Sarcopenia in Knee Osteoarthritis: a Systematic Review and Meta-analysis" for the Journal of Clinical Medicine. This study was a well-written article. I appreciate the steps in the method for systematic review and meta-analysis that are appropriately presented. However, there are some questions as follows

  1. Studies that used only low muscle mass could not be diagnosed sarcopenia. Could it be only pre-sarcopenia?

R: Thank you for the question. We have now included this point as possible limitation of our study, as follows:

Third, different definitions of sarcopenia were used in this systematic review, including definitions using only muscle mass assessment. In this regard, several societies indicated that low muscle mass identifies only a stage of pre-sarcopenia than sarcopenia itself, for which other measurements are needed, such as physical performance or muscle strength. [2,30] However, out findings reflect common situations in meta-research regarding sarcopenia, since, particularly in older works, only one domain of sarcopenia was assessed.”

  1. Who is the control for this study? Please clarify.

R: We are sincerely sorry for this inconvenience. We previously declared the controls only in the Introduction. Now, we included this important point in the Methods section, as follows:

“(iv) reporting data regarding controls defined as people without any evidence of knee OA.”

  1. The authors stated that the prevalence of sarcopenia in knee OA was 45.2% while in control was 31.2%. How can it result in the prevalence of sarcopenia in knee OA being more than two times higher than in the control group? Even OR = 2.07, can it be interpreted as presented?

R: Good point. We believe that to write that the prevalence of sarcopenia is doubled in knee OA patients compared to controls since the OR is =2.07. We have reported in the Discussion section this concept, as follows:

“Pooling the data of four selected studies, evaluating a sample of 7495 subjects, we clearly identified the prevalence of sarcopenia in people with knee OA (45.2%) resulting more than two times higher than in the control group (31.2%), since the OR was 2.07.” 

  1. In addition, after sensitivity analysis to decrease heterogeneity, the recalculated OR was lessened (1.88). The reviewer suggested adding this in the abstract.

R: Added.

Reviewer 3 Report

Thank you for inviting me to review this manuscript. It is a systematic review and meta-analysis addressing the Prevalence of Sarcopenia in Knee Osteoarthritis. This review is essential because, in the current time, providing direct evidence of an association between osteoarthritis and sarcopenia could help to better understand the progression and development of disability in older patients with knee osteoarthritis.

I have only one criticism: The authors included four cross-sectional studies, three conducted in Asia and one in Europe. My main concern is regarding the diagnostic criteria used to diagnose sarcopenia. The authors state that two included studies used only low muscle mass, and the others used a combination of low muscle mass and physical performance. Moreover, in the discussion, in lines 159-160, the authors stated that: (...) Furthermore, sarcopenia was defined as low muscle mass and/or physical performance as suggested by EWGSOP criteria (...). 

This statement needs to be better explained because the current international guidelines: European: EWGSOP2 and Asian: AWGS, define sarcopenia in this way:

* EWGSOP2: low muscle strength + low muscle mass  (1).

* AWGS: low muscle mass + low muscle strength or low physical performance(2).

Low muscle mass alone is not a sufficient criterion for diagnosing sarcopenia. Even the SDOC group (3, 4), in its latest position statement, excludes muscle mass from the definition of sarcopenia. Therefore, uncertainty exists regarding the sarcopenia status of these patients. 

This review provides conclusions on the prevalence of sarcopenia, even though it shows an association between knee osteoarthritis and low muscle mass (and not with sarcopenia per se).

References 

1.         Cruz-Jentoft AJ, Bahat G, Bauer J, Boirie Y, Bruyère O, Cederholm T,   et al. Sarcopenia: revised European consensus on definition and diagnosis. Age Ageing. 2019;48(1):16-31.

2.         Chen LK, Woo J, Assantachai P, Auyeung TW, Chou MY, Iijima K, et al. Asian Working Group for Sarcopenia: 2019 Consensus Update on Sarcopenia Diagnosis and Treatment. J Am Med Dir Assoc. 2020;21(3):300-7.e2.

3.         Bhasin S, Travison TG, Manini TM, Patel S, Pencina KM, Fielding RA, et al. Sarcopenia Definition: The Position Statements of the Sarcopenia Definition and Outcomes Consortium. J Am Geriatr Soc. 2020;68(7):1410-8.

4.         Cawthon PM, Manini T, Patel SM, Newman A, Travison T, Kiel DP, et al. Putative Cut-Points in Sarcopenia Components and Incident Adverse Health Outcomes: An SDOC Analysis. J Am Geriatr Soc. 2020;68(7):1429-37.

Author Response

R: We fully agree with this criticism. We have to better detail this relevant point, in the Limitations section, as follows:

“Third, different definitions of sarcopenia were used in this systematic review, including definitions using only muscle mass assessment. In this regard, several societies indi-cated that low muscle mass identifies only a stage of pre-sarcopenia than sarcopenia itself, for which other measurements are needed, such as physical performance or muscle strength. [2,30] However, out findings reflect common situations in me-ta-research regarding sarcopenia, since, particularly in older works, only one domain of sarcopenia was assessed.”

Reviewer 4 Report

To authors
This manuscript represents a major effort to prove the association between sarcopenia and knee OA. The paper provides important information; however, I think the following concerns should be addressed.

Major corrections

Regarding the number of included studies:

The number of the included studies in this manuscript are relatively few compared to a previous systematic review including article only from Ovid Medline and EMBASE databases till August 2018, the number of included studies was 15.

Amirthalingam H, Cicuttini FM, Wang Y, Chou L, Wluka AE, Hussain S. Association between sarcopenia and osteoarthritis-related knee structural changes: a systematic review. Osteoarthritis and Cartilage. 2019 Apr;27:S472.

Regarding the validity of results:

I think the data of the fourth study should be removed from assessment as it assessed different population (the only study assessed European patients with sample of 227 out of 7495 (3.02%).Therefore, limiting this systematic review and meta-analysis to Asian population would be more valid. Moreover, the 4th study comprised mainly of women (98.9%) compared to the other 3 studies.

Minor corrections

Line 19: the word studies is missing the letter S

Line 22: “sarcopenia in knee OA in patients was high” needs correction” in patients with knee OA” or omit the word in

Line 22: “affecting one person over two” correct to “affecting one every two persons’

Author Response

Major corrections

Regarding the number of included studies:

The number of the included studies in this manuscript are relatively few compared to a previous systematic review including article only from Ovid Medline and EMBASE databases till August 2018, the number of included studies was 15. Amirthalingam H, Cicuttini FM, Wang Y, Chou L, Wluka AE, Hussain S. Association between sarcopenia and osteoarthritis-related knee structural changes: a systematic review. Osteoarthritis and Cartilage. 2019 Apr;27:S472.

R: We sincerely thank the Reviewer 3 for indicating us this important paper. However, we are able to see only in form of a conference abstract. Therefore, we believe that these Authors have, probably, used different inclusion and exclusion criteria that finally lead to these different results in the study selection.

Regarding the validity of results:

I think the data of the fourth study should be removed from assessment as it assessed different population (the only study assessed European patients with sample of 227 out of 7495 (3.02%).Therefore, limiting this systematic review and meta-analysis to Asian population would be more valid. Moreover, the 4th study comprised mainly of women (98.9%) compared to the other 3 studies.

R: Good point. We understand this feeling, but we believe that it is a posteriori thought, whilst, according to a systematic approach to the literature we cannot exclude a study only because it includes people of other continents. Moreover, as reported, the study that was the most responsible of the high heterogeneity found is the study of Sung 2016 (females) further justifying the inclusion of the study of Udomsinprasert, 2020.

Minor corrections

Line 19: the word studies is missing the letter S

Line 22: “sarcopenia in knee OA in patients was high” needs correction” in patients with knee OA” or omit the word in

Line 22: “affecting one person over two” correct to “affecting one every two persons’

R: Thank you for the careful reading. We have corrected all these typos.

Round 2

Reviewer 3 Report

The authors maintain a narrow definition of sarcopenia. Even in the previous EWGSOP and AWGS publications (2010 and 2014, respectively)1,2, more than one criterion of muscle mass is needed to diagnose sarcopenia. The data analyzed refers only to the low muscle mass; therefore, discussing the prevalence of sarcopenia associated with knee OA is ambitious. Since It is unknown how many of these patients had sarcopenia.

A possible approach is to use the term Presarcopenia according to EWGSOP-2010.

1.       Cruz-Jentoft AJ, Baeyens JP, Bauer JM, Boirie Y, Cederholm T, Landi F, et al. Sarcopenia: European consensus on definition and diagnosis: Report of the European Working Group on Sarcopenia in Older People. Age Ageing. 2010;39(4):412-23.

2.         Chen LK, Liu LK, Woo J, Assantachai P, Auyeung TW, Bahyah KS, et al. Sarcopenia in Asia: consensus report of the Asian Working Group for Sarcopenia. J Am Med Dir Assoc. 2014;15(2):95-101.

Author Response

Thank you for this comment. We have further detailed this important information in the Limitations section, as suggested. 

Reviewer 4 Report

the authors responded adequately to the concerns aroused

Author Response

Thank you!